# The Influence of Social Support on the Job Involvement of Newly Hired Physical Education Teachers: A Study Based on SOR and COR Theories

**DOI:** 10.3390/bs15030271

**Published:** 2025-02-26

**Authors:** Lumin Liu, Tianpei Li, Duo Zhang, Hwang Jin

**Affiliations:** Physical Education, Jeonbuk National University, Jeonju 54896, Republic of Korea; liulumin1998@163.com (L.L.); tianpeili9611@163.com (T.L.); zhangduo9702@163.com (D.Z.)

**Keywords:** physical education teacher, social support, professional mission, job involvement, mediation

## Abstract

This study, grounded in Stimulus–Organism–Response (SOR) theory and Conservation of Resources (COR) theory, explores how social support impacts job involvement among newly hired physical education (PE) teachers, with a focus on the mediating role of professional mission. A survey was conducted with 238 new PE teachers, using scales for social support, professional mission, and job involvement. The results indicate the following: (1) Social support and its dimensions significantly and positively influence job involvement among new PE teachers; (2) Professional mission positively impacts job involvement and serves as a partial mediator in the relationship between social support and job involvement; (3) Different types of social support have varying effects on job involvement, with support utilization having the largest impact, followed by objective support, and finally subjective support.

## 1. Introduction

In the context of physical education, job involvement is defined as the enthusiasm, focus, and dedication that teachers exhibit in their professional roles. This involvement is closely related to the performance and satisfaction of physical education teachers, as high levels of job involvement typically lead to more effective teaching, higher student engagement, and greater job satisfaction for the teachers themselves ([14]). In China, the reform of basic education and the development of sports infrastructure have significantly impacted the job involvement of physical education teachers. In recent years, the Chinese government has increased its emphasis on physical education by implementing policy reforms and investing resources to improve school sports facilities and curricula ([29]). These changes have not only enhanced the quality of physical education but also provided teachers with better working environments and development opportunities, thereby promoting their job involvement and sense of professional achievement. Understanding these contextual factors is crucial for comprehensively analyzing the influence of social support on the job involvement of newly hired physical education teachers. Current research has identified several antecedents to job involvement, such as social support, positive emotions, self-efficacy, and job challenge ([18]; [23]). Job involvement has been shown to improve individual performance, work competence, and subjective well-being. To enhance teaching quality, increasing teachers’ job involvement is essential. As the demand for PE teachers rises due to China’s primary education reform and development of sports infrastructure, the number of newly hired teachers has also grown. However, these teachers often struggle to meet job demands due to limited early-career support ([15]). Newly hired teachers, often referred to as novice or beginning teachers, are generally those transitioning from student to teacher roles, facing stress and anxiety that can impact their work performance ([27]). This challenging period can significantly shape their commitment to teaching ([38]). Additionally, teachers with adverse early experiences often perform below expectations, possibly due to confidence loss and restricted professional development ([40]). In the long term, newly hired PE teachers represent an indispensable group in the evolution of the PE teaching workforce ([37]). Research indicates that individuals with high job involvement tend to be more energetic, proactive, and passionate about their work. Social support, encompassing the material and emotional support from close social connections, has been shown to significantly influence job involvement ([17]; [19]). In studying the job involvement of newly hired physical education teachers, social support is viewed as a multidimensional construct, primarily including emotional, informational, and instrumental support. Emotional support involves empathy, care, and understanding from colleagues, mentors, and administrators, helping new teachers feel valued and understood, thereby reducing stress and enhancing their sense of belonging. Informational support includes the provision of advice, guidance, and feedback, aiding teachers in adapting to their new roles and improving their confidence and competence. Instrumental support involves tangible assistance, such as resources, time, or help with specific tasks, ensuring that teachers have the necessary tools and conditions to perform their duties effectively. Understanding these dimensions of social support is crucial for exploring how they influence the professional mission and job involvement of newly hired physical education teachers. By fostering a comprehensively supportive environment, schools can enhance teachers’ engagement and satisfaction, ultimately improving educational outcomes. Good social support enables individuals to feel appreciated and supported, reducing negative emotions, enhancing psychological well-being, and improving job quality ([39]). This study aims to explore the impact of social support on job involvement among newly hired PE teachers, with the objective of enhancing job involvement to improve the quality of PE education and expedite China’s progress as a global sports leader.

### 1.1. Theoretical Foundations and Research Hypotheses

#### 1.1.1. SOR (Stimuli–Organism–Response) Theory

In 1935, Skinner hypothesized that the environment could influence human behavior by affecting responses to stimuli. Mehrabian later expanded this into the SOR model, which explains how environmental factors impact human behavior ([21]). This model, originally applied to consumer behavior studies ([26]), has since been used to analyze public health events, learning engagement, and more ([25]; [34]). The SOR model includes three elements: stimulus, organism, and response, representing antecedent variables, mediating variables, and outcome variables, respectively ([7]). “Stimulus” refers to factors affecting cognition or emotion ([31]), “Organism” refers to the psychological or cognitive changes in response to stimuli ([3]), and “Response” is the behavioral outcome after processing emotions and cognition ([4]). In the context of the physical education industry, the SOR model provides a valuable framework for understanding how social support influences job involvement among newly hired physical education teachers. Social support is viewed as a stimulus that can enhance job involvement by providing emotional, informational, and instrumental resources ([3]). Information-processing theory suggests that one’s environment shapes behaviors and attitudes through a complex processing system, where extensive psychological activity occurs between stimulus and response ([4]). Thus, in the SOR model, “Organism” reflects the individual’s psychological changes after stimulus processing. For physical education teachers, professional mission serves as this “Organism,” representing a value-driven cognitive state that motivates them to engage actively in their work ([10]). This intrinsic motivation leads to experiencing joy and fulfillment, aligning well with the SOR model’s framework. Job involvement, as the “Response,” indicates the degree of identification and engagement employees have with their work ([35]). In the physical education sector, highly involved teachers are more likely to contribute positively to student outcomes and organizational performance ([2]). This study establishes connections among social support, professional mission, and job involvement using the SOR model’s “stimulus–organism–response” logic, highlighting its applicability in enhancing the professional experiences of physical education teachers.

#### 1.1.2. COR (Conservation of Resources) Theory

COR Theory, or Conservation of Resources theory, highlights resources as crucial for enhancing job performance and supporting organizational growth. Initially conceptualized by [12] ([12]), COR theory focuses on resource loss and gain, proposing “loss spirals” and “gain spirals”. Loss spirals indicate that individuals lacking resources experience continued stress from resource depletion, while gain spirals suggest those with ample resources tend to preserve and acquire additional resources. Hobfoll identifies relationship and individual resources as foundational for job involvement, with relationship resources (e.g., organizational support, including leadership, peer support, and cultural support) as crucial for job engagement, and individual resources (e.g., optimism and self-efficacy) enhancing value recognition and a sense of mission ([13]). In this study, social support is considered an external motivator falling within the relationship resources category, while professional mission serves as an intrinsic motivator, heightening work motivation and engagement among newly hired PE teachers ([8]). Utilizing SOR and COR theories, this study aims to elucidate the mechanisms behind job involvement through external stimuli, psychological changes, relationship resources, and individual resources.

#### 1.1.3. Direct Relationship Between Social Support and Job Involvement

Social support refers to the material and emotional assistance from individuals with close relationships, categorized into subjective support, objective support, and support utilization ([33]). This classification is widely applied in research in the field. Social exchange theory suggests that all human actions and social relationships are reward-oriented, adhering to reciprocal exchange principles. Organizations provide conditions to meet employees’ needs, strengthening employee-organization bonds and increasing job involvement ([4]). Positive psychology has introduced job involvement as a mental health indicator, with social support proven to positively impact job involvement ([6]). Research shows that social support allows employees to perform tasks with ease, offering emotional support under stress, thus boosting productivity ([30]). Studies confirm a positive correlation between social support and job involvement across various sectors, including trade, banking, telecommunications, and healthcare, as well as among Portuguese nurses ([23]). Given this established link between social support and job involvement, we propose Hypothesis H1(a, b, c): Social support (objective support, subjective support, and utilization of support) positively impacts job involvement among newly hired PE teachers.

#### 1.1.4. Mediating Role of Professional Mission

Neo-classical theory defines “professional mission” as a sense of purpose that transcends self-interest, linking one’s career to broader life goals driven by altruism and personal values ([10]). In the context of newly hired physical education teachers, professional mission emphasizes the pursuit of meaningful educational impact and personal growth. Studies have shown that a strong professional mission enhances job involvement and subjective well-being, fostering a deeper connection to one’s work ([9]; [22]). Self-Determination Theory (SDT) (Figure 1) categorizes motivation into amotivation, extrinsic motivation, and intrinsic motivation. According to SDT, autonomous motivation strengthens internal motivation, with professional mission being driven by a genuine love for teaching and contributing to student development ([28]). This intrinsic motivation is crucial for new physical education teachers as they adapt to their roles and seek fulfillment in their careers. Recent studies have linked professional mission to increased job involvement, suggesting that a higher sense of mission enhances attachment to the organization and job satisfaction ([1]; [2]). For physical education teachers, this means that a strong professional mission can lead to greater engagement and effectiveness in their teaching roles. According to SOR theory, external stimuli such as social support lead to cognitive and psychological changes, which in turn result in behavioral adjustments. In this framework, social support acts as a stimulus that influences the professional mission (organism), ultimately enhancing job involvement (response). This process underscores the importance of fostering a supportive environment for new teachers, enabling them to develop a strong professional mission and achieve higher levels of job involvement. Professional mission, as a cognitive and internal motivator, positively influences job involvement, leading to Hypothesis H2: Professional mission positively affects job involvement among newly hired PE teachers.

There is a significant relationship between social support and a sense of professional mission. According to Xie Baoguo, social situational factors influence an individual’s sense of professional mission, with organizational or social support exerting a positive impact on this sense of mission ([33]). Bunderson’s study on managers found that the formation of a sense of professional mission is influenced by both intrinsic professional cognition factors and extrinsic environmental factors ([36]). Social support is an essential factor for new teachers in recognizing their “teacher identity” and in forming a sense of professional mission. Based on existing research, we propose Hypothesis H3 (a, b, c): Social support (objective support, subjective support, and utilization of support) has a positive effect on the sense of professional mission among newly employed physical education teachers.

According to the literature cited above, there is some research support for the correlation between social support, sense of professional mission, and work engagement. The Stimulus–Organism–Response (SOR) theory suggests that external environmental stimuli lead to changes in an individual’s cognition and psychology, which in turn alter their behavior. This theoretical framework implies that cognitive and psychological changes play a mediating role between external stimuli and behavioral responses. Additionally, information processing theory indicates that there is considerable psychological activity between stimulus and response, which also partially supports the mediating role of the sense of professional mission. Therefore, we hypothesize that the effect of social support on work engagement may be mediated by the sense of professional mission; that is, the social support received by newly employed physical education teachers could stimulate a sense of professional mission, which would, in turn, lead to greater enthusiasm and dedication to their work, thereby enhancing their level of work engagement. Accordingly, we therefore propose Hypothesis H4 (a, b, c): A sense of professional mission serves as a mediating factor between social support (objective support, subjective support, and utilization of support) and work engagement for newly employed teachers.

In summary, this study employs the SOR model and conservation of resources theory to construct a research model in which social support acts as the antecedent variable, the sense of professional mission as the mediating variable, and work engagement as the outcome variable, thereby modeling the effect of social support on work engagement among newly employed physical education teachers (see Figure 2).

### 1.2. Research Subjects and Tools

#### 1.2.1. Research Subjects

Simple random sampling was usedand the questionnaire was collected for a total of one month. The subjects of this study were newly hired physical education teachers with less than or equal to 3 years of experience in primary, junior high, and high schools in the Shandong, Jilin, Liaoning, and Shanxi provinces. A total of 300 paper questionnaires were distributed, with 269 returned, giving a response rate of 89.7%. After data cleaning, 238 valid questionnaires were retained, resulting in an effective rate of 79.3%. This research protocol was approved by the Ethics Committee of Binzhou University (BZU—20231101) and complies with the ethical standards of the 1964 Helsinki Declaration and its later amendments. All participants signed informed consent forms.

#### 1.2.2. Research Tools

##### Social Support Rating Scale

The Social Support Rating Scale, developed by Xiao Shuiyuan, was used to measure social support among newly hired physical education teachers. This scale consists of three dimensions and includes 10 items scored on a 4-point scale. The total score is the sum of the dimensions, with a higher score indicating a greater level of social support. In this study, the Cronbach’s α coefficient of the scale was 0.852, indicating good internal consistency. The structural validity indices were as follows: χ^2^/df = 2.25, GFI = 0.944, AGFI = 0.904, RMSEA = 0.073, CFI = 0.950, NFI = 0.915, IFI = 0.951, suggesting good structural validity.

##### Professional Mission Scale

The Professional Mission Scale, developed by Zhang Chunyu, was used to measure the professional mission of newly hired physical education teachers. This scale has three dimensions and includes 11 items scored on a 5-point scale, with a higher score indicating a stronger sense of professional mission. In this study, the Cronbach’s α coefficient of the scale was 0.847, indicating good internal consistency. The structural validity indices were as follows: χ^2^/df = 2.132, GFI = 0.937, AGFI = 0.899, RMSEA = 0.069, CFI = 0.932, NFI = 0.881, IFI = 0.933, suggesting good structural validity.

##### Job Involvement Scale

The Job Involvement Scale, developed by [29] ([29]), was used to measure job involvement among newly hired physical education teachers. This scale has three dimensions and includes 17 items scored on a 6-point scale, with a higher score indicating greater job involvement. In this study, the Cronbach’s α coefficient of the scale was 0.888, indicating good internal consistency. The structural validity indices were as follows: χ^2^/df = 2.13, GFI = 0.937, AGFI = 0.899, RMSEA = 0.069, CFI = 0.932, NFI = 0.881, IFI = 0.933, suggesting good structural validity.

### 1.3. Analysis Process

In the data analysis step of this study, SPSS 26.0 and AMOS 26.0 software were used for statistical processing. First, descriptive statistical analysis was performed on the collected data to understand the basic characteristics of the sample. Then, regression analysis was used to test the mediation model to evaluate the relationship between social support, occupational mission, and job engagement. Through the mediation effect analysis, we used the Bootstrap method to test the significance of the mediation effect to ensure the robustness of the results. All analyses were performed at the significance level of 0.05.

## 2. Results Analysis

### 2.1. Demographic Variables Statistics

As shown in Table 1, a total of 238 participants were included in this study, comprising 134 males (56.3%) and 104 females (43.7%). Regarding education level, there were 118 undergraduate students (49.6%), 103 master’s degree holders (43.3%), and 17 doctoral degree holders (7.1%). In terms of years of work experience, 98 participants had 1 year (41.2%), 96 had 2 years (40.3%), and 44 had 3 years (18.5%). Overall, the study covers physical education teachers of different genders, education levels, and work experience, offering a good level of representativeness.

### 2.2. Common Method Bias Test

This study used Harman’s single-factor test to examine common method bias. All measured items were subjected to factor analysis, and the results showed that there were eight factors with eigenvalues greater than 1. The largest factor explained 24.02% of the variance, which is below the standard threshold of 40%. This indicates that common method bias is unlikely to significantly affect the data analysis, allowing for further investigation.

### 2.3. Descriptive Statistics and Correlation Analysis

As shown in Table 2, social support is positively correlated with objective support, subjective support, support utilization, professional mission, and job involvement. Objective support is positively correlated with subjective support, support utilization, professional mission, and job involvement. Subjective support is positively correlated with support utilization, professional mission, and job involvement. Support utilization is positively correlated with professional mission and job involvement. Additionally, professional mission is positively correlated with job involvement. These results provide the preliminary support for the study.

### 2.4. The Impact of Social Support on Job Involvement of Newly Hired Physical Education Teachers

Following the mediation testing procedure recommended by [32] ([32]), a regression analysis was conducted with social support as the independent variable and job involvement as the dependent variable. The results indicate that social support has a significant positive effect on job involvement of newly hired physical education teachers (β = 0.386, t = 3.130, *p* = 0.002). This suggests that the higher the level of social support, the greater the job involvement of these teachers. Further analysis examined whether the dimensions of social support—objective support, subjective support, and support utilization—separately influence the job involvement of newly hired physical education teachers. Gender and educational level were included as control variables, and nine regression analyses were conducted, with job involvement and professional mission as dependent variables (Table 3). Additionally, the variance inflation factor (VIF) values in the model ranged from 1 to 1.106, indicating no multicollinearity issues between variables. Detailed analysis is as follows:

(1)When job involvement was the dependent variable:

In Model 1, gender and educational level did not significantly affect job involvement of newly hired physical education teachers (F = 0.239, *p* > 0.05).

In Model 2, objective support had a significant positive effect on job involvement (β = 0.359, *p* < 0.001), supporting Hypothesis H1a.

In Model 3, subjective support had a significant positive effect on job involvement (β = 0.324, *p* < 0.001), supporting Hypothesis H1b.

In Model 4, support utilization had a significant positive effect on job involvement (β = 0.445, *p* < 0.001), supporting Hypothesis H1c.

In Model 5, professional mission had a significant positive effect on job involvement (β = 0.419, *p* < 0.001), supporting Hypothesis H2.

(2)When professional mission was the dependent variable:

In Model 6, gender and educational level did not significantly affect professional mission (F = 0.496, *p* > 0.05).

In Model 7, objective support had a significant positive effect on professional mission (β = 0.370, *p* < 0.001), supporting Hypothesis H3a.

In Model 8, subjective support had a significant positive effect on professional mission (β = 0.484, *p* < 0.001), supporting Hypothesis H3b.

In Model 9, support utilization had a significant positive effect on professional mission (β = 0.309, *p* < 0.001), supporting Hypothesis H3c.

### 2.5. Testing for Mediation Effects

Currently, there are two main methods for testing mediation effects with multiple independent variables. One approach is to include multiple independent variables in a single model, but this method carries the risk that the predictive effects of the variables may offset each other. The other approach is to create separate models for each independent variable and analyze each model individually, which is more widely used. In this study, the second approach was adopted. The mediation effect of professional mission was tested using the Bootstrap method with the SPSS macro command provided by Preacher and Hayes.

As can be seen in Table 4, the total effect, direct effect, and indirect effect values of objective support are 2.569, 1.684, and 0.885, respectively. The direct effect accounts for 65.551% of the total effect, and the indirect effect accounts for 34.449%. The 95% confidence intervals (1.713, 3.425), (0.812, 2.556), and (0.462, 1.419) do not include 0, indicating that professional mission partially mediates the relationship between objective support and work engagement among newly recruited physical education teachers. Thus, Hypothesis H4a is supported. For subjective support, the total effect, direct effect, and indirect effect values are 1.122, 0.547, and 0.575, respectively. The direct effect accounts for 48.752% of the total effect, and the indirect effect accounts for 51.125%. The 95% confidence intervals (0.702, 1.541), (0.092, 1.002), and (0.327, 0.839) do not include 0, indicating that professional mission partially mediates the relationship between subjective support and work engagement among newly recruited physical education teachers. Thus, Hypothesis H4b is supported. For the utilization of support, the total effect, direct effect, and indirect effect values are 3.145, 2.456, and 0.689, respectively. The direct effect accounts for 78.092% of the total effect, and the indirect effect accounts for 21.908%. The 95% confidence intervals (2.326, 3.964), (1.642, 3.269), and (0.347, 1.102) do not include 0, indicating that professional mission partially mediates the relationship between the utilization of support and work engagement among newly recruited physical education teachers. Thus, Hypothesis H4c is supported.

## 3. Discussion

### 3.1. The Direct Influence of Social Support on Work Engagement of Newly Recruited Physical Education Teachers

The results of this study show that social support and its various dimensions significantly positively predict the work engagement of newly recruited physical education teachers, consistent with the findings of [28] ([28]). This indicates that the stronger the social support an individual perceives, the higher their level of work engagement. According to Self-Determination Theory, people naturally tend toward psychological growth and integration, which motivates them to learn, master, and connect with others ([5]). Although individuals can make free choices based on environmental information and personal needs, they are also influenced by external environmental factors ([11]). When individuals perceive support and help from family, colleagues, and friends, they are more likely to engage actively in their work, showing higher levels of work engagement. Compared to veteran teachers, newly recruited physical education teachers are often recent graduates, generally younger and less experienced, and may not yet be fully adapted to the transition from school life to social life. Thus, when new physical education teachers receive support and assistance from friends around them, it undoubtedly boosts their confidence in their work, encouraging them to engage in their roles with a more positive attitude. Furthermore, as noted earlier, “encountering difficulties in the early stages of employment can severely limit the confidence and career development of new teachers”. In the long term, providing necessary support and assistance to newly recruited physical education teachers can strengthen their lifelong commitment to teaching and bolster their confidence in the profession ([24]). From the perspective of Conservation of Resources Theory, the level of work engagement is associated with the resource ecology. Individuals with abundant resources can objectively understand work requirements and challenges, while those lacking resources tend to exaggerate them. Social support, as examined in this study, falls under relational resources. Previous research suggests that when relational resources are scarce, individuals may reduce their emotional attachment to the organization and view colleagues and leaders as sources of external pressure. If individuals remain in this negative emotional state for extended periods, they may experience anxiety and depression, reducing their level of work engagement. Therefore, from the perspective of resource conservation theory, providing necessary support and assistance to newly recruited physical education teachers can enhance their trust in and sense of belonging to the organization, satisfying emotional and work needs, and mitigating the mental and physical distress caused by work, enabling them to fully engage in their roles.

### 3.2. The Mediating Role of Professional Mission

The results of this study indicate that the professional mission of newly recruited physical education teachers partially mediates the relationship between social support and work engagement. On the one hand, social support directly influences work engagement; on the other hand, it has an indirect effect on work engagement through its positive influence on professional mission. The attitude change model suggests that factors influencing attitude change can be categorized into internal and external factors ([11]). Work engagement, as a work attitude of newly recruited physical education teachers, is affected not only by external environmental stimuli but also by an internal sense of mission. Interpreting this further through the lens of resource conservation theory, the richness of employees’ work resources significantly affects their level of work engagement; employees with abundant relational and individual resources tend to have higher levels of work engagement. In this study, social support belongs to relational resources, and professional mission belongs to individual resources. Thus, when newly recruited physical education teachers have rich social support and a high level of professional mission, their work engagement level is likely to increase.

Firstly, from the perspective of the support and assistance perceived by new teachers, good social support can provide new teachers with more emotional care. When teachers feel supported, cared for, and respected by friends and colleagues, it not only motivates them to complete their teaching tasks diligently but also enhances their affirmation and recognition of their profession and role, further developing a strong professional mission. Therefore, social support, as an external environmental stimulus, can guide newly recruited physical education teachers to view lifelong dedication to education as a means of realizing personal value, merging personal value with the educational profession, and fostering a strong professional mission. This view aligns with the research of [24] ([24]), [20] ([20]), and [16] ([16]), which also indicate that high levels of social support have a significant positive effect in the workplace. Additionally, in China, teachers are often associated with a spirit of dedication, emphasizing selfless dedication to students, which contributes to a generally high level of professional mission among Chinese teachers ([39]). Teachers with a high sense of professional mission can comprehend the significance of the educational profession more deeply during their teaching careers. As a lasting internal driving force, a professional mission enables newly recruited physical education teachers to transcend a surface-level understanding of education, closely linking personal values and goals with their teaching careers, fostering an intrinsic identification with their work and a positive work attitude. This not only eliminates various negative emotions but also increases their level of work engagement. According to the SOR (Stimulus–Organism–Response) theory, external environmental stimuli and cognitive psychological changes work together to influence behavioral changes. In this study, this theory is specifically reflected in the way that when newly recruited physical education teachers perceive support and assistance from their surroundings, it strengthens their professional mission, which in turn serves as an internal driving force that stimulates a higher level of work engagement. Therefore, professional mission plays a mediating role in the relationship between social support and work engagement among newly recruited physical education teachers.

### 3.3. Comparison of the Effects of Different Types of Social Support on Work Engagement of Newly Recruited Physical Education Teachers

The results of this study indicate that different types of social support (objective support, subjective support, and utilization of support) have varying effects on the work engagement of newly recruited physical education teachers. A comparison reveals that the utilization of support has the greatest total effect on work engagement, followed by objective support, with subjective support having the smallest effect. Further analysis of the mediation models shows that the indirect effect of objective support on work engagement is greater than that of the utilization of support, while the direct effect is smaller than the latter. Generally, objective support refers to visible or tangible support and assistance, including material aid, and subjective support is the emotional support that individuals perceive. In this study, the objective support perceived by newly recruited physical education teachers is more meaningful than subjective support, possibly because these teachers are new to the profession and, upon receiving tangible support and assistance, experience emotional changes that solidify their commitment to education, thereby increasing the indirect effect of objective support on work engagement. During the survey, we observed variations in how different newly recruited physical education teachers utilized support. When facing the same support and assistance, some teachers felt supported, while others did not.

## 4. Conclusions and Implications

### 4.1. Conclusions

From the research described above, the following conclusions are drawn: (1) Social support positively influences the work engagement of newly recruited physical education teachers, with objective support, subjective support, and utilization of support all having positive effects on work engagement, and utilization of support having the largest effect. (2) Professional mission positively affects work engagement among newly recruited physical education teachers and partially mediates the relationships between objective support, subjective support, and utilization of support and work engagement.

### 4.2. Implications

Research implications: (1) Social support directly promotes the work engagement of newly recruited physical education teachers. Therefore, schools, parents, and society should provide sufficient support to encourage teachers to engage fully in their work. (2) Professional mission is an important factor influencing the impact of social support on work engagement among new teachers. In terms of mechanisms, self-efficacy can directly stimulate work engagement among new teachers and further foster stronger self-efficacy when they receive support from schools, parents, and society, thereby enhancing work engagement. Thus, professional mission can directly stimulate work engagement and also partially mediate the relationship between social support and work engagement. (3) This study shows that among the dimensions of social support, the total effect values rank in descending order as utilization of support, objective support, and subjective support. Therefore, when providing support and assistance to newly recruited physical education teachers, particular attention should be given to how teachers utilize the support. Although the levels of support and assistance received by different teachers are generally similar, there is variability in how much they can actually make use of it. Thus, the degree of utilization of support can be considered an important factor for assessment.

## 5. Research Limitations

This study has several limitations that need to be addressed in future research. First, while the SOR (Stimulus–Organism–Response) and COR (Conservation of Resources) theories have been widely applied in psychology, their use in studying the relationship between social support and work engagement among teachers remains limited. This study employed SOR and COR theories to construct a model explaining work engagement in newly recruited physical education teachers, and also incorporated motivation theory. However, other theories related to work engagement, such as the Job Demands-Resources (JD-R) model and Herzberg’s Two-Factor Theory, were not explored in this study. Future researchers could use different theoretical frameworks to further explain the relationship between social support and work engagement among teachers.

Second, due to time and geographical constraints, the sample coverage in this study is limited. Most of the newly recruited physical education teachers surveyed are from regions like Shandong, Jilin, Liaoning, and Shaanxi, with a lack of representation from southern and western areas. This limitation may affect the generalizability of the study’s findings. Expanding the sample to include teachers from different regions in future studies would help ensure broader applicability of the conclusions.

Lastly, this study’s participants are restricted to newly recruited physical education teachers, while teachers from other disciplines were not included. Teachers from different disciplines may exhibit unique characteristics. Future research should encompass multiple subject areas, investigating the relationships between social support, vocational sense of mission, and work engagement among teachers from various fields. This would provide more targeted support and assistance to teachers, helping them engage more fully in their work.

## Figures and Tables

**Figure 1 behavsci-15-00271-f001:**
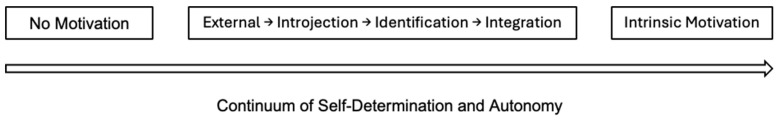
Continuum of motivation change.

**Figure 2 behavsci-15-00271-f002:**
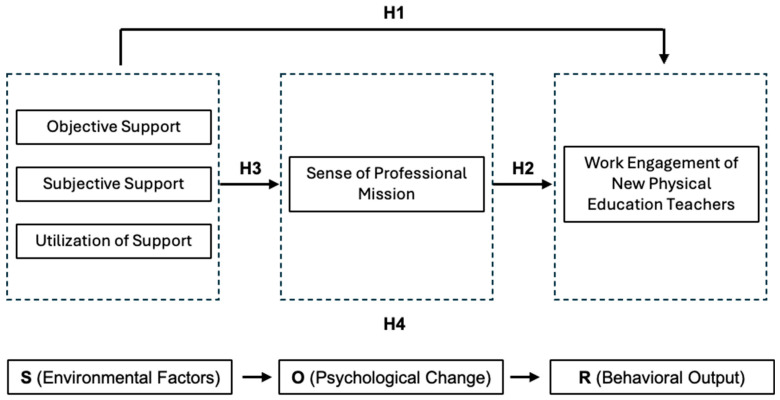
Stimulus–Organism–Response (SOR) theory analytical model.

**Table 1 behavsci-15-00271-t001:** Demographic information of the sample.

Category	Option	Frequency	Percentage (%)
Gender	Male	134	56.3
	Female	104	43.7
Education	Bachelor’s	118	49.6
	Master’s	103	43.3
	Doctorate	17	7.1
Years of Work Experience	1 year	98	41.2
	2 years	96	40.3
	3 years	44	18.5
Total		238	100

**Table 2 behavsci-15-00271-t002:** Means, standard deviations, and correlation coefficients between variables (*N* = 238).

Variable	M	SD	1	2	3	4	5	6
1	3.389	1.030	1					
2	2.323	0.927	0.653 **	1				
3	5.005	1.438	0.870 **	0.710 **	1			
4	2.690	0.932	0.635 **	0.546 **	0.656 **	1		
5	3.529	0.866	0.471 **	0.372 **	0.484 **	0.309 **	1	
6	3.876	1.170	0.200 **	0.359 **	0.324 **	0.442 **	0.420 **	1

Note: 1 = social support, 2 = objective support, 3 = subjective support, 4 = utilization of support, 5 = sense of professional mission. 6 = work engagement; ** *p* < 0.01.

**Table 3 behavsci-15-00271-t003:** Test of direct effects.

Variable	Work Engagement					Sense of Professional Mission			
	Model 1	Model 2	Model 3	Model 4	Model 5	Model 6	Model 7	Model 8	Model 9
Gender	0.44	0.007	0.013	0.037	0.018	0.061	0.023	0.014	0.057
Education Level	−0.009	0.008	0.017	0.038	0.000	−0.020	−0.003	0.018	0.012
Objective Support		0.359 ***					0.370 ***		
Subjective Support			0.324 ***					0.484 ***	
Utilization of Support				0.445 ***					0.309 ***
Sense of Professional Mission					0.419 ***				
R^2^	0.002	0.129	0.106	0.198	0.177	0.004	0.139	0.235	0.099
ΔR^2^	−0.006	0.118	0.094	0.188	0.166	−0.004	0.128	0.225	0.087
F	0.239	11.5599 ***	9.2079 ***	19.2429 ***	16.7349 ***	0.496	12.601 ***	23.900 ***	8.5384 ***

Note: *** *p* < 0.001.

**Table 4 behavsci-15-00271-t004:** Analysis of the effects of objective support, subjective support, and utilization of support on work engagement.

Variable	Effect Type	Effect	SE	Bias-Corrected 95% CI	
				Lower	Upper
Objective Support	Total Effect	2.569	0.435	1.713	3.425
	Direct Effect	1.684	0.443	0.812	2.556
	Indirect Effect	0.885	0.244	0.462	1.419
Subjective Support	Total Effect	1.122	0.213	0.702	1.541
	Direct Effect	0.547	0.231	0.092	1.002
	Indirect Effect	0.575	0.130	0.327	0.839
Utilization of Support	Total Effect	3.145	0.416	2.326	3.964
	Direct Effect	2.456	0.413	1.642	3.269
	Indirect Effect	0.689	0.195	0.347	1.102

## Data Availability

The data that support the findings of this study are available from the corresponding author H.J. (Hwang Jin, email: jeanh@jbnu.ac.kr), upon reasonable request.

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
