# Peer review of "The Influence of Social Support on the Job Involvement of Newly Hired Physical Education Teachers: A Study Based on SOR and COR Theories"

_behavsci, 2025, doi:10.3390/bs15030271_

Round 1

Reviewer 1 Report

Comments and Suggestions for Authors

The abstract you present has a good focus and is clear about the objectives and main results of the study. I should point out that, although it is mentioned that a sample of 300 new physical education (PE) teachers was used, it would be useful to specify more details about the population (e.g. the country or region where the study was conducted, if relevant). In addition, while academic audiences are likely to be familiar with SOR and COR theories, it would be useful to briefly mention how these theories apply to the context of PE teachers, to provide more context for the reader who may not be fully familiar.

Regarding the introduction, in some paragraphs the sentences are long and complex, which can make comprehension difficult. I think it would be good to simplify the sentences or break them into shorter sentences. Also, in the first paragraph the definition of ‘work engagement’ is important, but it could be clearer if a brief clarification is provided on the implications of this concept in the context of physical education (PE). For example, how does work engagement relate to the performance and satisfaction of PE teachers specifically? The mention of ‘China's primary education reform and development of sports infrastructure’ is relevant but could be briefly expanded upon to give readers unfamiliar with the Chinese context a better understanding of the importance of this reform and the growth of sports infrastructure.

In the second paragraph of the introduction, the transition between the general explanation of work involvement and the situation of new FE teachers could be smoother. You can use a sentence linking the two concepts.

In section 1.1.1, the concept of SOR theory is well introduced, but it would be useful to elaborate a bit more on how it applies specifically to FE teachers. To what extent does the stimulus (social support) trigger a response (job involvement) in FE teachers?

In section 1.1.2, the use of COR theory is adequate, but may benefit from better integration with SOR theory. It could be specified how relational resources, such as social support, help new teachers to ‘conserve’ internal resources such as motivation and resilience, which is crucial for maintaining a high level of work engagement. In addition, it would be useful to provide concrete examples of the types of ‘resources’ that new teachers may gain or lose and how this impacts their performance.

The hypotheses are well formulated, but it would be useful to more clearly justify the relationship between the variables.

I would like to point out that a section devoted to Methodology is missing from the article, this would be point 2. Methodology is critical for readers to understand how the study was conducted and how the data were collected and analysed. This section should include:

1. research design: is it a quantitative, qualitative or mixed study? What type of design was used (experimental, correlational, etc.)?

2.   Participants: Although it is mentioned that the sample consisted of 300 newly recruited physical education teachers, it would be useful to provide more details on the participant selection process, such as the type of sampling used (random, convenience, etc.) and any inclusion or exclusion criteria.

3.   Instruments and measurement: Although the scales used to measure social support, professional mission and work engagement are mentioned, it would be important to describe in more detail how these instruments were administered, as well as their validity and reliability.

4.   Procedure: This should include information on how the data collection was carried out, such as whether it was online, in person, how long the data was collected, etc.

As for the Results section, the demographic data are well presented, and the breakdown of gender, educational level and years of work experience helps to contextualise the sample. It would be advisable to explain in more detail how the sample was obtained (whether random sampling was used, convenience sampling, etc.). This can improve the external validity of the study. In addition, I think it might be useful to include the average age of the participants or more details about the distribution of their work experience, as these factors may influence the results of subsequent analyses.

The use of Harman's one-factor test to assess common method bias is good practice and is well explained. Although the analysis shows that common method bias is not a major problem, it would be useful to briefly discuss the implications of this result and why this particular method was chosen.

As for Descriptive Statistics and Correlation Analysis, the correlation results are clearly presented, and the relationships between the main variables are well described. I think it would be useful to add a brief interpretation of the highest or most significant correlation values. This would help readers to understand the significance of these relationships before delving into the regression analysis.

In The Impact of Social Support on Job Involvement of Newly Hired Physical Education Teachers, regression is an appropriate technique for examining the impact of independent variables on the dependent variable (job involvement). The results are clear and provide evidence to support the hypothesis that social support influences job involvement. It would be useful to include a more detailed analysis of control effects, such as gender and educational level. Although it is mentioned that these factors are not significant, a deeper interpretation of why they had no effect could be relevant.

Mediation analysis using the Bootstrap method is appropriate for exploring the indirect effects of the variables, and the results are presented in full, with confidence intervals for each effect. Although the indirect and direct effects are clearly presented, it is important to reflect on how the mediation results might have practical implications for the field of physical education. How can social support programmes improve teachers' work engagement?

In terms of discussion, the findings of the study are clearly and coherently presented, highlighting how social support positively influences the work engagement of new physical education teachers. This part is well supported by theories such as self-determination theory and resource conservation theory, which provides a solid theoretical framework. In addition, the analysis of mediation through the professional mission provides a clear and useful explanation of how social support affects the work engagement of new teachers. The application of resource conservation theory to link relational resources to personal resources (professional mission) is interesting and well-founded. It would be useful to discuss any possible confounding variables that might have affected the relationship between social support, professional mission and work engagement. For example, factors such as teachers' intrinsic motivation or perception of job stability could influence these links. The comparison between different types of social support (objective, subjective and utilisation of support) is a strong aspect of the discussion, as it helps to understand variations in how these types of support affect teachers' work engagement. The results suggest that objective support has a more significant effect than subjective support, which is an important observation. In the last part of this section, it is mentioned that newly recruited teachers vary in how they use the support they receive. It would be interesting to explore further the reasons behind this variability: are there personal or contextual factors (such as the type of support received, organisational culture or individual teacher characteristics) that explain why some teachers do not feel the support that others perceive? This could open up an interesting area for future research.

Regarding the conclusion section, the conclusions are well organised and present a clear and concise summary of the most important findings of the study. Each conclusion is directly related to the objectives of the study and the results obtained. I must point out that a more detailed analysis of how cultural and regional differences may influence the results of the study and the practical implications would be a valuable addition.

In the references section, some references have minor inconsistencies in formatting, especially in the use of capital letters in article and book titles. References 31 and 33 appear to repeat information similar to reference 22 (Work as a calling by Duffy et al.). Some names are spelled inconsistently (e.g. Hobfol, S. E. should be Hobfoll, S. E.). I consider that the authors should make a thorough revision of this section.

I believe that, although the article represents a valuable contribution to the study of the influence of social support on the work engagement of newly recruited physical education teachers, several revisions are needed to optimise its clarity, methodological rigour and overall quality. I believe that the authors need to address the issues identified, such as elaborating the theoretical frameworks more fully, improving the coherence and structure of the introduction, and incorporating a detailed methodological section. Also, a more thorough analysis of the results, discussion of possible confounding variables, and careful refinement of the references are essential steps to strengthen the work and move towards publication.

Author Response

Dear reviewer:

We feel great thanks for your professional review work on our article.As you are concerned,there are several problems that need to be addressed.According to your nice suggestions,we have made extensive corrections are listed below.

Comments1:{The abstract you present has a good focus and is clear about the objectives and main results of the study. I should point out that, although it is mentioned that a sample of 300 new physical education (PE) teachers was used, it would be useful to specify more details about the population (e.g. the country or region where the study was conducted, if relevant). In addition, while academic audiences are likely to be familiar with SOR and COR theories, it would be useful to briefly mention how these theories apply to the context of PE teachers, to provide more context for the reader who may not be fully familiar.}

Response1:{Thanks for your suggestion.We have redlined the selected areas for the study in section 1.2.1. In addition, in section 1.1.1 we have restated how the theoretical model can be applied to physical education teachers.}

Comments2 :{Regarding the introduction, in some paragraphs the sentences are long and complex, which can make comprehension difficult. I think it would be good to simplify the sentences or break them into shorter sentences. Also, in the first paragraph the definition of ‘work engagement’ is important, but it could be clearer if a brief clarification is provided on the implications of this concept in the context of physical education (PE). For example, how does work engagement relate to the performance and satisfaction of PE teachers specifically? The mention of ‘China's primary education reform and development of sports infrastructure’ is relevant but could be briefly expanded upon to give readers unfamiliar with the Chinese context a better understanding of the importance of this reform and the growth of sports infrastructure.}

Response2:{Thanks for your suggestion.Based on expert advice, we have revised the introduction on the first page to make it more accessible and added a section on physical education in China. In addition, we also reinterpreted the relationship between work engagement and physical education teachers' satisfaction.}

Comments3:{In the second paragraph of the introduction, the transition between the general explanation of work involvement and the situation of new FE teachers could be smoother. You can use a sentence linking the two concepts.}

Response3:{Thanks for your suggestion.Based on expert advice, we modified the second paragraph of the introduction to make the sentence flow better.}

Comments4:{In section 1.1.1, the concept of SOR theory is well introduced, but it would be useful to elaborate a bit more on how it applies specifically to FE teachers. To what extent does the stimulus (social support) trigger a response (job involvement) in FE teachers?}

Response4:{Thanks for your suggestion.Based on the experts' recommendations, we refined how the SOR theory can be applied specifically to physical education teachers in section 1.1.1, detailing the mechanism of the role of stimulation (social support) and physical education teachers' work engagement.}

Comments5:{I would like to point out that a section devoted to Methodology is missing from the article, this would be point 2. Methodology is critical for readers to understand how the study was conducted and how the data were collected and analysed.}

Response5:{Thanks for your suggestion.Based on the experts' suggestions, we have added the type of sampling in this paper in section 1.2.1, explaining how the questionnaire was collected and how long it took, and describing the areas from which the 300 new physical education teachers were selected and their inclusion criteria. In addition, in 1.2.2 we describe in detail the scales selected for this paper and explain that each scale contains several dimensions also make a statement about the reliability and validity.}

Comments6:{As for Descriptive Statistics and Correlation Analysis, the correlation results are clearly presented, and the relationships between the main variables are well described. I think it would be useful to add a brief interpretation of the highest or most significant correlation values. This would help readers to understand the significance of these relationships before delving into the regression analysis.}

Response6:{Dear experts, part 2.3 does not explain the highest correlation value because the highest correlation in this paper is between subjective support and social support, and subjective support is under the social support dimension, plus this paper explores the relationship between social support, sense of occupational mission and work engagement so it is not explained. We apologize to the experts for any inconvenience this may cause.}

Comments7:{Mediation analysis using the Bootstrap method is appropriate for exploring the indirect effects of the variables, and the results are presented in full, with confidence intervals for each effect. Although the indirect and direct effects are clearly presented, it is important to reflect on how the mediation results might have practical implications for the field of physical education. How can social support programmes improve teachers' work engagement?}

Response7:{Thanks for your suggestion.In line with the experts' recommendations, we explain in more detail in section 1.1.4 how the mediator variable works between social support and work engagement, and how the mediator variable can improve PE teachers' work engagement in practice to.}

Comments8:{Regarding the conclusion section, the conclusions are well organised and present a clear and concise summary of the most important findings of the study. Each conclusion is directly related to the objectives of the study and the results obtained. I must point out that a more detailed analysis of how cultural and regional differences may influence the results of the study and the practical implications would be a valuable addition.}

Response8:{Dear experts, we did not point out regional and cultural differences in the conclusion section because the sample was selected from China and belonged to the same ethnic group. We apologize for any inconvenience caused to the experts.}

Comments9:{In the references section, some references have minor inconsistencies in formatting, especially in the use of capital letters in article and book titles. References 31 and 33 appear to repeat information similar to reference 22 (Work as a calling by Duffy et al.). Some names are spelled inconsistently (e.g. Hobfol, S. E. should be Hobfoll, S. E.). I consider that the authors should make a thorough revision of this section.}

Response9:{Based on the experts' recommendations, we have further standardized the format of the references.}

We tried our best to improve the manuscript and made some changes marked in red in revised paper which will not influence the content and framework of the paper.We appreciate foe Editors and Reviewer’s warm work earnestly,and hope the correction will meet with approval.Once again,thank you very much for your comments and suggestions.

Reviewer 2 Report

Comments and Suggestions for Authors

1. It is stated in the abstract that 300 new PE teachers took part in the study, but later it is stated that 238 participants were included in the study (final sample).

2. In the conclusion section it is stated that "demographic factors such as gender and educational background do not impact the professional mission and work engagement of newly recruited teachers" but there is no other reference in the paper concerning the role of demographic factors, not even in the abstract. If demographic factors were supposed to be one of the variables, then they should have been analyzed at the first part of the manuscript.

3. Title of the journals at the Reference section are not in italics, as it is adressed by APA.

4. In section 1.1.1 the SOR model is explained, but at the start of the paragraph  SOR is written only abbreviated and not as a whole (Stimulus-Organism-Response Theory), which makes it difficult for the reader to understand what it is about.

5. It could be useful if at the theoretical background more details were given concerning the physical education (PE) profession in China, as well as previous research that investigated newly hired PE teachers and their relation to the study's variables (social support, professional mission and job involvement).

6. The types of social support should have been briefly analyzed at the first part of the manuscript, as it is not clear what each dimension is about. 

7. Conclusions could have drawn from the discussion, but not repeat the discussion part. Its is important to provide an argument following the results. Moreover, conclusion no 1 (demographic factors) as it was mentioned earlier in comment no 2 were not stated clearly at the start of the article that were one of the variables.

Author Response

Dear reviewer:

We feel great thanks for your professional review work on our article.As you are concerned,there are several problems that need to be addressed.According to your nice suggestions,we have made extensive corrections are listed below.

Comments1:{It is stated in the abstract that 300 new PE teachers took part in the study, but later it is stated that 238 participants were included in the study (final sample).}

Response1:{Thanks for your suggestion.Based on the experts' recommendations, we modified the sample size in the summary.}

Comments2:{In the conclusion section it is stated that "demographic factors such as gender and educational background do not impact the professional mission and work engagement of newly recruited teachers" but there is no other reference in the paper concerning the role of demographic factors, not even in the abstract. If demographic factors were supposed to be one of the variables, then they should have been analyzed at the first part of the manuscript.}

Response2:{Thanks for your suggestion.In accordance with the recommendations of the experts, we have made changes to 4.1 Conclusions by deleting and modifying the conclusions on demographic factors.}

Comments3:{Title of the journals at the Reference section are not in italics, as it is adressed by APA.}

Response3:{Thanks for your suggestion.Based on the recommendations of the experts, we have further standardized the format of the references.}

Comments4:{ In section 1.1.1 the SOR model is explained, but at the start of the paragraph  SOR is written only abbreviated and not as a whole (Stimulus-Organism-Response Theory), which makes it difficult for the reader to understand what it is about.}

Response4::{Thanks for your suggestion.Based on the comments of the experts, we have modified the acronym of the SOR theory in Section 1.1.1 in an attempt to make it more understandable.}

Comments5:{It could be useful if at the theoretical background more details were given concerning the physical education (PE) profession in China, as well as previous research that investigated newly hired PE teachers and their relation to the study's variables (social support, professional mission and job involvement).}

Response5:{Thanks for your suggestion.Following the advice of the experts, we provide a more detailed background on physical education in China in the introduction section, as well as an explanation of the relationships between the variables.}

Comments6:{The types of social support should have been briefly analyzed at the first part of the manuscript, as it is not clear what each dimension is about.}

Response6::{Thanks for your suggestion.In line with expert advice, we have detailed aspects of social support in the Introduction section.}

Comments7:{Conclusions could have drawn from the discussion, but not repeat the discussion part. Its is important to provide an argument following the results. Moreover, conclusion no 1 (demographic factors) as it was mentioned earlier in comment no 2 were not stated clearly at the start of the article that were one of the variables.}

Response7::{Thanks for your suggestion.In accordance with the Expert's recommendations, we have deleted the conclusion on demographics in 4.1.}

We tried our best to improve the manuscript and made some changes marked in red in revised paper which will not influence the content and framework of the paper.We appreciate foe Editors and Reviewer’s warm work earnestly,and hope the correction will meet with approval.Once again,thank you very much for your comments and suggestions.

Round 2

Reviewer 1 Report

Comments and Suggestions for Authors

After reviewing the modifications made to the manuscript ‘The Influence of Social Support on Job Involvement of Newly Hired Physical Education Teachers: A Study Based on SOR and COR Theories’, I believe that they have adequately addressed the observations raised during the review process.grading.